# Smoke Back-Layering Phenomenon under the Combined External Wind and Stack Effects in a Staircase

Man Li *, Lingling Wang, Junya Chen, Zhenrong Mu and Suqi Liu

College of Civil Engineering, Huaqiao University, Xiamen 362021, China; 14759@hqu.edu.cn (L.W.); chenjunya@126.com (J.C.); Zhenrong_Mu@163.com (Z.M.); chiggaliu647@163.com (S.L.)
* Correspondence: liman312@hqu.edu.cn; Tel.: +86-13856079896

**Abstract:** The external wind can change smoke movement patterns inside the staircase and affect smoke exhaust efficiency. This paper analyzes the smoke back-layering phenomenon in the staircase with open stair doors below the fire floor. The effect of the open stair door location and the heat release rate of fires and external wind velocities on smoke movement patterns are investigated numerically. The external wind ranges from 0–5.5 m/s. At 0 m/s, the smoke back-layering phenomenon driven by pressure difference can be found in the staircase with all stair doors closed. With the increasing wind velocity, four smoke behaviors are identified: upward moving smoke, first downward then upward moving smoke, downward moving smoke, and no smoke. Results show that the back-layering distance is mainly influenced by the external wind and heat release rate of fires. Correlations are modified and used to predict the longest back-layering distance with the first downward then upward moving smoke. This helps with arranging the smoke detectors inside a staircase and the fire safety design of high-rise buildings.

**Keywords:** external wind; stack effect; staircase; open stair door; smoke back-layering



## 1. Introduction

With the accelerating process of urbanization in the world, high-rise buildings and super tall buildings have increased dramatically. Fire safety of high-rise buildings has attracted extensive attention in recent years due to the continuous disastrous high-rise building fires, such as the New CCTV tower fire in 2009 and the Grenfell Tower fire in 2017. Compared to normal compartment fires, the fire behavior in high-rise buildings has some special features, including (1) the specific smoke transport due to complex building structures [1] and (2) a windy environment around the buildings.

During fires, many vertical shafts in high-rise buildings, such as emergency stairwells, elevator shafts, and ventilating ducts, will become the potential paths of smoke spread. The smoke will spread from the fire floor to the other floors, causing fatalities far away from the fire origin [2]. The stack effect is the main driven force of smoke movement inside the staircase, which is caused by the pressure difference generated from the density difference of hot gases and cold airs inside and outside of the building [3]. Many characteristic parameters of the smoke movement driven by stack effect were studied, such as the vertical temperature and pressure distribution [4,5], rising velocity of the smoke propagation [6,7], and the neutral plane position [8]. Additionally, correlations to predict deflection characteristics of the tilted flame under stack effect were established [9].

Untenable conditions occurred quicker when fires were wind-driven [10], and the wind environment around the building cannot be neglected. Most scholars have researched wind effects on compartment fires, including the height and temperature distribution of the external flame under cross wind [11], external flame projection probability under assisting wind environment [12], and temperature evolution inside a compartment under opposing wind environment [13]. Wind effects on high-rise building fires are more complicated

owing to the interaction of the wind and stack effect, resulting in special smoke movement phenomenon and flame behaviors in a room connected to a shaft. Ji et al. [14] found that, when the wind velocity was lower than a critical value, the flame tilted towards the outdoor at early stage and then tilted towards the shaft eventually due to the stack effect. A non-dimensional number was also proposed to determine the flame tilting direction. Later, Ji et al. [15] proposed a quantitative model to predict the rise time of plume front under the combined effects of external wind and stack effect. However, the fire source was always located on the first floor in their studies. Then, they could observe that the smoke flowed upward inside the staircase when stack effect overcame the external wind and flowed outside the fire room when the stack effect could not overcome the external wind. However, the fire may occur on any floor in real fires. When the fire source was located on the ninth floor, Li et al. [2] found the smoke moved downward and flowed out from the lower opening with a positive pressure ventilation in the staircase. As they focused on the effect of pressurized air leakage on smoke prevention, the characteristics of the downward moving smoke and its influencing factors were not studied further. The smoke moved downward inside the staircase when the thermal buoyancy was not larger enough to overcome the inertia force of the wind, the so-called "smoke back-layering phenomenon". The smoke back-layering distance was defined as the length of the reserving smoke flow downstream when the wind velocity was larger than the critical velocity [16], which was influenced by the heat release rate (HRR) of fires and the open stair door below the fire floor. The stair doors were kept open to improve the natural ventilation in some high-rise residences sometimes. This poses new challenges in determining the smoke management in fire, as the smoke movement pattern could affect smoke exhaust efficiency. Therefore, it is important to improve the understanding of smoke behavior and its spread in the staircase.

In previous publications, most studies focused on characteristics of the upward moving smoke [3–9]. Some studied flame behaviors under the competition between the thermal buoyancy of fires and the inertia force of external wind with the fire on the first floor [14,15]. Then, there was no smoke back-layering phenomenon occurring inside the staircase. Compared to the previous publication, this paper is mainly focused on the smoke back-layering phenomenon and the characteristic parameter of back-layering distance under the wind environment. In the current study, the fire source was located on the eighth floor. The effect of the open stair door below the fire source on the smoke movement was first researched. Then, the smoke back-layering phenomenon under the combined effect of wind and stack effect was studied. Finally, a correlation was formulated to predict the smoke back-layering distance.

## 2. Numerical Modeling

FDS (fire dynamics simulator software) is regarded as a practical tool for simulating fire-induced thermodynamics and fire science. The model has been subjected to numerous validations and calibrations in high-rise building fire simulations [17–19]. FDS (version 6.7.3) is employed in the current work. FDS solves the Navier–Stokes equations with a low-Mach number formulation for thermally driven flow. It includes both a LES (Large Eddy Simulation) model and a DNS (Direct Numerical Simulation) model. LES is adopted as the turbulence model. Deardorff turbulent viscosity is used in FDS 6 by default,

$$\mu_{LES}/\rho = C_v \Delta \sqrt{k_{sgs}} \qquad (1)$$

where $C_v = 0.1$. Based on scale similarity, the subgrid scale (sgs) kinetic energy is taken from an algebraic relationship [20]. The LES filter width is taken as the geometric mean of the local mesh spacing in each direction, $\Delta = (\delta x \delta y \delta z)^{1/3}$. In an LES calculation, the convective heat flux can be expressed as

$$q_c'' = h(T_g - T_w) \qquad (2)$$

where $h$ is the convective heat transfer coefficient, which is based on a combination of forced and natural convection correlations:

$$h = \max\left[C(T_g - T_w)^{1/3}, \frac{k}{L}Nu, \frac{k}{\delta n/2}\right] \tag{3}$$

where $C$ is an empirical coefficient for natural convection, $L$ is a characteristic length related to the size of the physical obstruction, $k$ is the thermal conductivity of the gas, $N_u$ is the Nusselt number, and $\delta n$ is the gas phase cell size. The last term of the $h$ formula aims to ensure that the LES convective heat flux converges to the DNS heat flux at fine resolutions. The Radiative Transfer Equation (RTE) is solved using the Finite Volume Method (FVM). A radiation fraction of 0.35 is prescribed as a lower bound in order to limit the uncertainties in the radiation calculation induced by uncertainties in the temperature field. Heat losses to the walls are calculated by solving the 1-D Fourier's equation for conduction [18,20]. More details can be seen in [20].

The simulation model is a high-rise building with 12 stories, as shown in Figure 1, consisting of the staircase, atrium, and compartment. The building is 36.6 m high. The ground floor is 3.6 m high, and the other floors are 3.0 m high. The cross-sections of staircase, atrium and compartment are, respectively, 4.5 m (L) × 3 m (W), 2.4 m (L) × 2.4 m (W), and 2.4 m (L) × 2.4 m (W). Three doors (D1–D3) connect the staircase, atrium, compartment, and surroundings. The sizes of the doors are all 1.8 (H) × 1.2 m (W). The window located on the back of the staircase is 3 m high by 2 m wide. The doors (D1–D3) on the eighth floor (8F) and window are always open. The doors (D1–D3) on the first (1F), third (3F), fifth (5F), and seventh (7F) floors remain open, respectively, and other doors remain closed. The internal lining of the building is specified as "CONCRETE", and its density, specific heat, and conductivity are 2200 kg/m$^3$, 0.88 kJ/(kg K), and 1.2 W/(m K) [20]. The ambient temperature was 20 °C. The external wind is provided by a wind screen and blows into the staircase from the window on the top floor. Wind velocity ranges from 0 to 5.5 m/s. The effects of open stair doors below the fire floor, heat release rate of fires, and the external wind velocity are considered.

A vertical column of 23 thermocouples are located at the centerline of the staircase, as shown in Figure 1. It is mainly used to determine the smoke back-layering time and distance. The fire locates on 8F and at the center of the compartment. The reaction was specified as propane gas fire. The HRRs are designed as 2, 3, and 4 MW. 123 tests are simulated, and the simulation time are more than 400 s. More details are shown in Table 1.

**Table 1.** Simulation details.

| Fire Floor | HRR/MW | Open Stair Doors (D1–D3) | Wind Velocity (m/s) |
|---|---|---|---|
| 8th | 4 | 7th | 0, 0.5, 1.5, 2.0 2.5, 3, 3.5, 4, 4.5, 5, 5.5 |
| | | 5th | 0, 0.5, 1.5, 2.0 2.5, 3, 3.5, 4, 4.5, 5 |
| | | 3rd | 0, 0.5, 1.5, 2.0 2.5, 3, 3.5, 4, 4.5, 5 |
| | | 1st | 0, 0.5, 1.5, 2.0 2.5, 3, 3.5, 4, 4.5, 5 |
| | | None | 0 |
| | 3 | 7th | 0, 0.5, 1.5, 2.0 2.5, 3, 3.5, 4, 4.5, 5, 5.5 |
| | | 5th | 0, 0.5, 1.5, 2.0 2.5, 3, 3.5, 4, 4.5, 5 |
| | | 3rd | 0, 0.5, 1.5, 2.0 2.5, 3, 3.5, 4, 4.5, 5 |
| | | 1st | 0, 0.5, 1.5, 2.0 2.5, 3, 3.5, 4, 4.5, 5 |
| | 2 | 7th | 0, 0.5, 1.5, 2.0 2.5, 3, 3.5, 4, 4.5, 5 |
| | | 5th | 0, 0.5, 1.5, 2.0 2.5, 3, 3.5, 4, 4.5, 5 |
| | | 3rd | 0, 0.5, 1.5, 2.0 2.5, 3, 3.5, 4, 4.5, 5 |
| | | 1st | 0, 0.5, 1.5, 2.0 2.5, 3, 3.5, 4, 4.5, 5 |

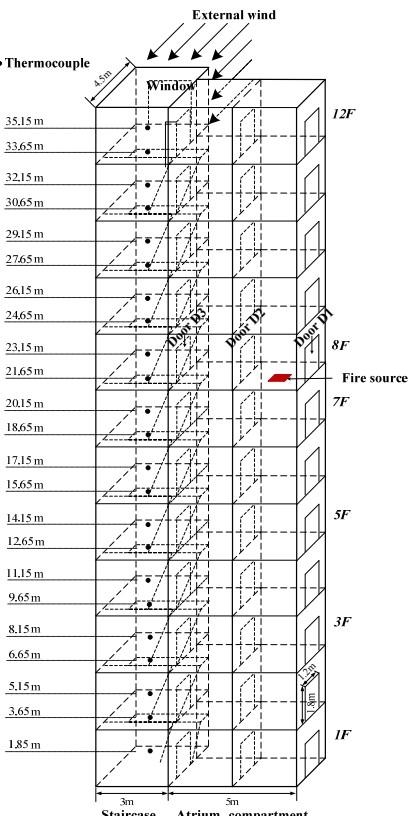

**Figure 1.** Schematic view of model building.

In the FDS user's guide, a $D^*/\delta x$ criterion has been used for assessing the grid resolution. $\delta x$ is the nominal size of a mesh cell, and $D^*$ is a characteristic fire diameter. $D^*$ is calculated by:

$$D^* = \left(\frac{Q}{\rho_\infty c_p T_\infty \sqrt{g}}\right)^{2/5} \tag{4}$$

It is recommended by McGrattan et al. [20] that the value of $D^* = \delta x$ should be in the range of 4–16. Then, the size of the finest mesh for a 2–4 MW fire was calculated to be between 0.07 and 0.4 m. Obviously, the finer grid will better reflect the heat flow field in detail, but it is also time-consuming. The mesh grid size was chosen to be 0.167 m in this study, which corresponds with previous studies [2,21,22]. To further verify the accuracy of the LES model, the vertical temperature distributions inside the staircase without external wind are compared with previous research [9,23], as shown in Figure 2. Qi et al. [24] researched the similarity laws between the full and small scales model and obtained the modified Froude–Stanton modeling. It can be expressed as

$$Q_m = \left(\frac{H_m{}^2 R_{tf}\rho_f}{H_f{}^2 R_{tm}\rho_m}\right) Q_f \tag{5}$$

$Q$ is the HRR, $H$ is the height of the staircase, $R_t$ is thermal resistance of the staircase, and $\rho$ is gas density. The subscript '$f$' and '$m$' represent, respectively, the full and model scale parameters. Based on the Froude–Stanton modeling, HRR of the 1/6 scale model is about 800 kW in the full scale model in [9]. HRR is about 360 kW in the full scale experiments in [23] and 2 MW in the current study. As shown in Figure 2, the temperature rise distribution performs the same variation tendency, and it increased with the increasing HRR. Then, the simulation results are believed to be accurate to some extent.

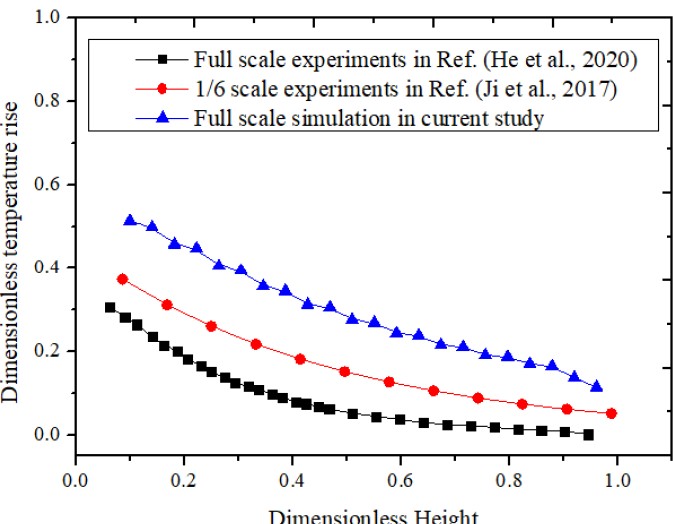

**Figure 2.** Vertical temperature distributions inside the staircase at 0 m/s [9,23].

## 3. Results and Discussion

### 3.1. Effect of the Open Stair Door below the Fire Source on Smoke Movement Patterns

In actual fires, the staircase is the main evacuation route for humans to escape where the stair doors will be opened above or below the fire floor. When the stair doors opened above the fire floor, the strength of the stack effect was influenced by the height between the open stair door floor and fire floor, which was studied by Ji et al. [9]. When the stair doors opened below the fire floor at the first, third, fifth, and seventh floors, the smoke movement patterns are similar to each other at the simulation in the current work, taking the case with open stair door on the first floor as an example.

At 0 m/s, Figure 3 shows the temperature distributions in the staircase with closed stair doors and an open stair door on the first floor. To be conservative, this study selects 300–400 s from the start of the fire as the steady state condition for averaging. It can be seen that the temperature distributions are similar above 25 m (9F) and different between 17–25 m (about 6F–9F). For the case with open stair door on the first floor, as shown in Figure 3a, temperatures are about 100–150 °C in the region far from the stair door at the height of 21.6–25 m and much lower than those in the case with stair doors closed. For the case with stair doors closed, as shown in Figure 3b, there are temperature rises at the height of 17–21.6 m (below 8F). Temperatures are at about 20 °C at the height below 17 m in both cases. Figure 4 shows the velocity vector graphs. Combining Figures 3a and 4a, it can be concluded that the air flows into the staircase from the open stair door on the first floor and moves upward. When the smoke enters the staircase from the door D3 on the eighth floor, the smoke tends to attach the sidewall and moves upward. Additionally, the air entrainment and convective heat transfer are enhanced by the airflow. Therefore, temperatures are quite low at the height of 21.6–25 m and then increase owing to the accumulation of the smoke below the stair trends at the height of 25–27 m. Combining Figures 3b and 4b, it can be concluded that when the smoke enters the staircase from the eighth floor, most of the smoke moves upward driven by the stack effect, and some smoke moves downward driven by pressure difference. Figure 5 shows the pressure differences inside and outside the staircase. $P_o$ is denoted as the air pressure at the ground outside the staircase. $P_A$ and $P_B$ can be expressed as:

$$P_A = P_o - \rho_a g h_1 - 18 \tag{6}$$

$$P_B = P_o - \rho_a g h_2 - 24 \tag{7}$$

$$\Delta P = P_A - P_B = \rho_a g (h_2 - h_1) + 6 \tag{8}$$

where $\rho_a$ is the air density (kg/m$^3$), $g$ is the gravity acceleration (m/s$^2$). If the pressure difference between *A* and *B* is a positive value, the smoke will move downward until a pressure equilibrium occurrs. Additionally, it can be seen from Figure 5 that the pressure differences inside and outside the staircase increase with the increasing height. The pressure difference below the fire floor in the case with closed stair doors is much lower than that with an open stair door on the first floor.

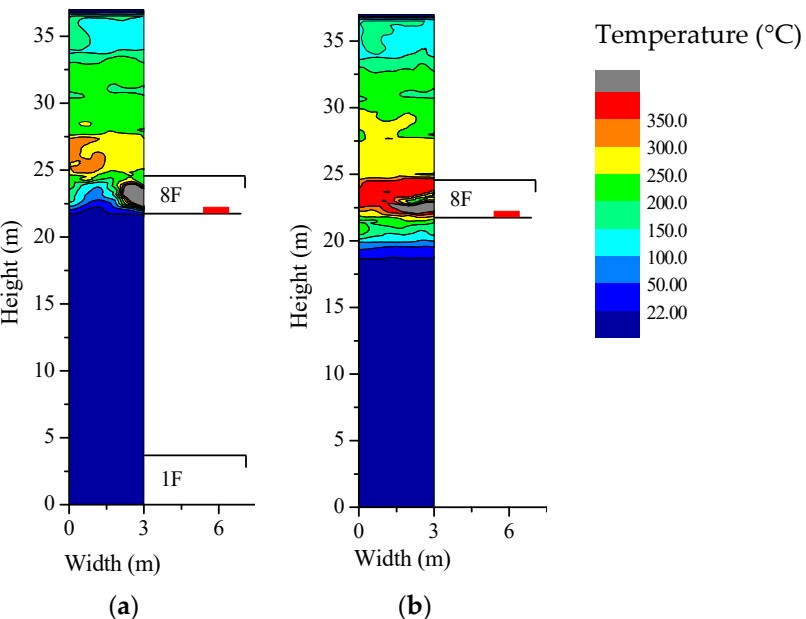

**Figure 3.** Temperature distributions in the staircase at 0 m/s. (**a**) Stair door opened on the first floor. (**b**) Stair doors closed.

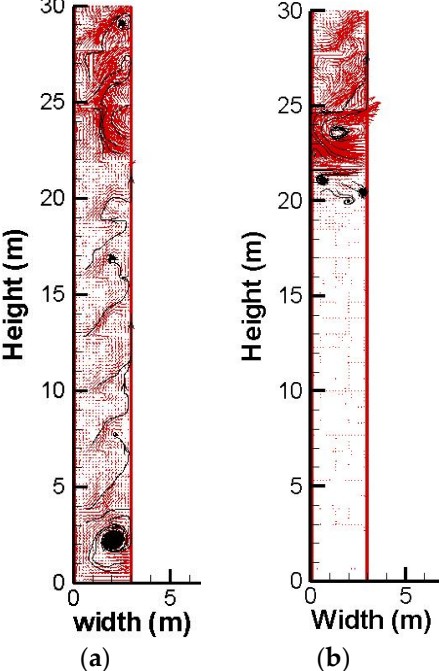

**Figure 4.** Velocity vector graphs in the staircase at 0 m/s. (**a**) Stair door opened on the first floor. (**b**) Stair doors closed.

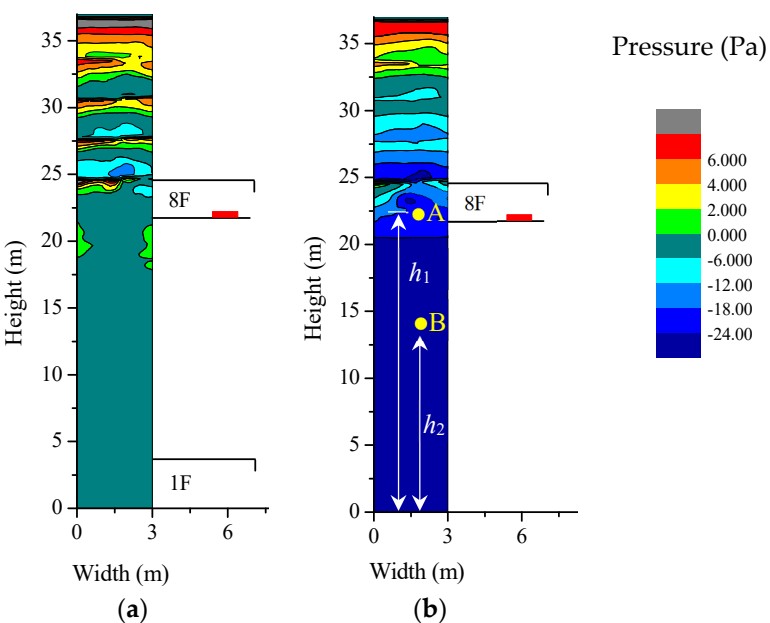

**Figure 5.** Pressure difference distributions in the staircase at 0 m/s. (**a**) Stair door opened on the first floor. (**b**) Stair doors closed.

### 3.2. Wind Effect on the Smoke Movement with Open Stair Door below the Fire Source

Smoke behaviors with different open stair door location under wind condition are similar at the simulation in the current work, taking cases with open stair door on the first floor as examples.

Figure 6 presents time history of vertical temperature profiles at the centerline of the staircase with open stair door on the first floor. At 0–3 m/s, as shown in Figure 6a, temperatures at different height above the fire floor increase with time sequentially and fluctuates around a constant value at the steady state. Temperatures at the height below the fire floor keep the ambient temperature all the time.

At 3.5 m/s, as shown in Figure 6b, most of the temperatures at different height increase with time. However, for the thermocouple at the height of at 21.65 m, there is a sudden temperature increment at about 20 s, which reaches a peak value of 50 °C approximately at 50 s and then decays rapidly. Figure 7 shows the velocity vector graphs inside the staircase at 3.5 m/s. Combining Figures 6b and 7a, it can be concluded that, when the smoke begins to enter the staircase, some smoke moves upward, and some moves downward. The longest back-layering distance occurred at about 50 s. Combining Figures 6b and 7b, we can see that the smoke and the airflow from the open stair door on the first floor move upward, driven by the stack effect eventually. This is because the thermal buoyancy of the smoke is too low to overcome the wind force when it is starts to enter the staircase. Some of the smoke is pushed to move downward, and the smoke back-layering phenomenon occurs. Then, the thermal buoyancy of the smoke and HRR of the fire increase with time and overcome the wind force. The smoke and airflow from the open stair door move upward, driven by the stack effect finally. Figure 8 presents temperature distributions in the atrium and fire compartment at 3.5 m/s. It can be seen that the flame tilts to the door D1 at 30 s and stands vertically at 52 s, then tilts to the door D3 at 70 s. Combining Figures 7b and 8b, it can be concluded that the time when the longest back-layering distance appears is consistent with the critical transition time of the flame tilting direction.

At 4 m/s, as shown in Figure 6c, temperatures keep the ambient temperature all the time at the height above the fire floor. However, temperatures increase sequentially first, then decay to a low value at the height below the fire floor. Figure 9 presents velocity vector graphs inside the staircase at 4 m/s. Combining Figures 6c and 9, it can be concluded that some smoke flows into the staircase and is pushed to move downward by the external wind. There will be a sudden temperature rise when the downward moving smoke front

reaches the thermocouple at a given height. The smoke flows out from the open stair door below the fire floor after a sudden temperature rise at 1.85 m. Additionally, Figure 10 shows temperature distributions in the atrium and fire compartment. The flame tilts to the door D3 at 120 s when the smoke moves downward. It is indicated that the flame tilting direction keeps toward the door D3 as the smoke flows out from the open stair door. Smoke behavior is directly related to the flame tilt direction. The theoretical model to determine the flame tilting direction established by Ji et al. [14] is also valid to predict the smoke behavior.

At 4.5 m/s, as shown in Figure 6d, temperatures inside the staircase always keep the ambient temperature. The thermal buoyancy of the smoke cannot overcome the wind force. The whole staircase is smoke-free and safe.

Vertical temperature distributions in the staircase under different wind velocities are shown in Figure 11. As shown in Figure 11, when the smoke overcomes the wind force and moves upward, temperatures decrease with the increasing height above the fire floor, and the external wind almost has no influence on the smoke temperatures.

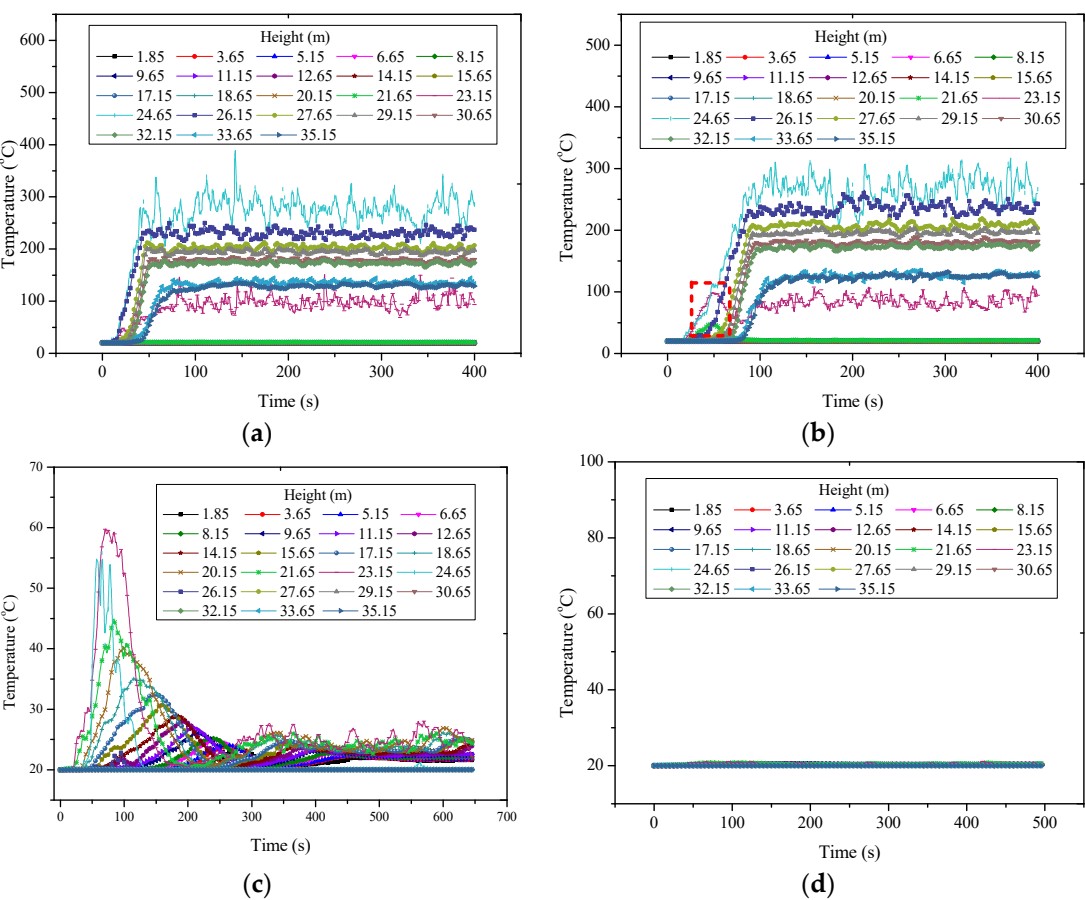

**Figure 6.** Vertical temperature profile versus time with open stair door on the 1st floor of 3 MW: (**a**) 0 m/s, (**b**) 3.5 m/s, (**c**) 4 m/s, and (**d**) 4.5 m/s.

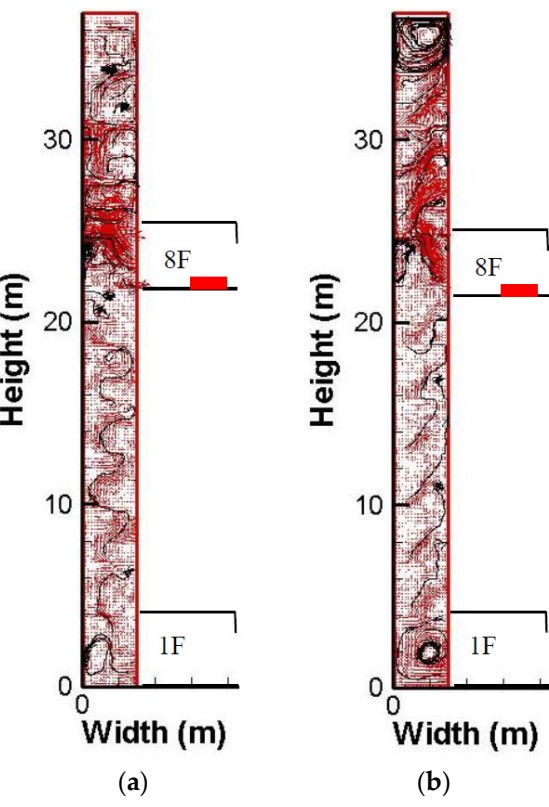

**Figure 7.** Velocity vector graphs at 3.5 m/s: (**a**) 52 s and (**b**) 390 s.

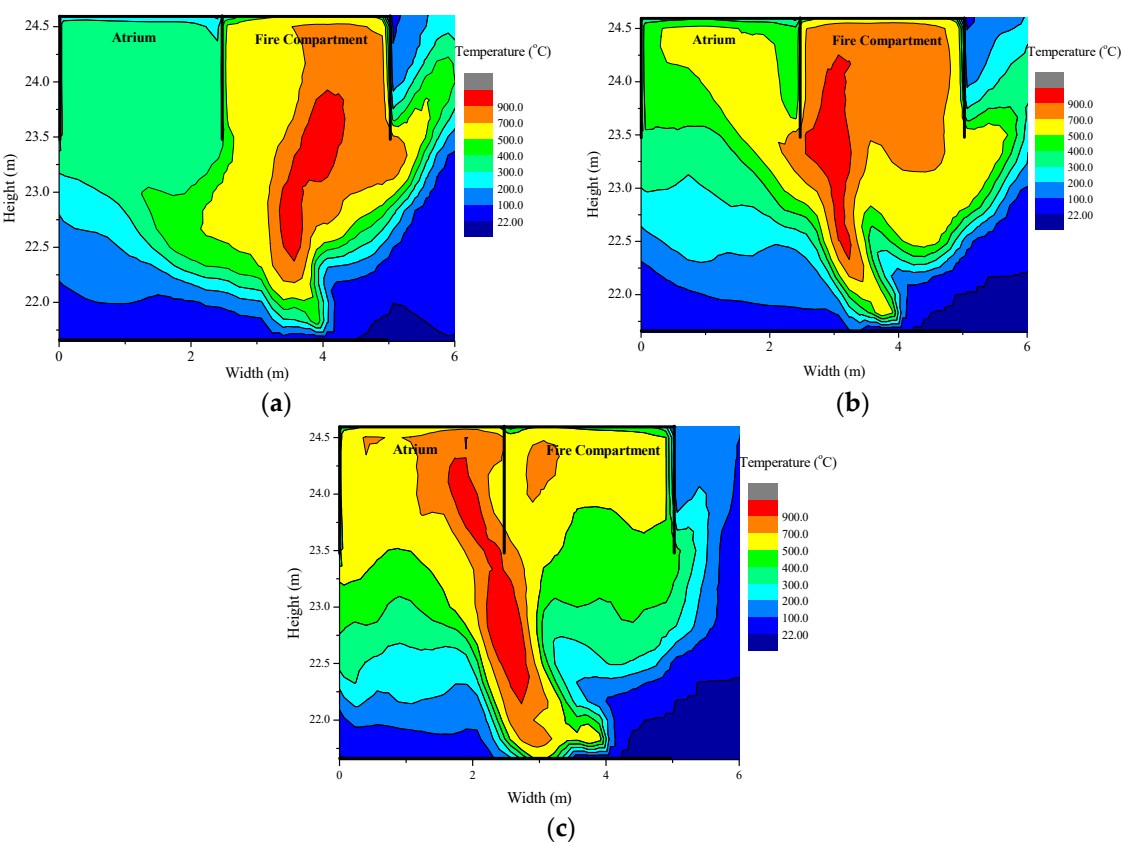

**Figure 8.** Temperature distributions in the atrium and fire compartment at 3.5 m/s: (**a**) 30 s, (**b**) 52 s, and (**c**) 70 s.

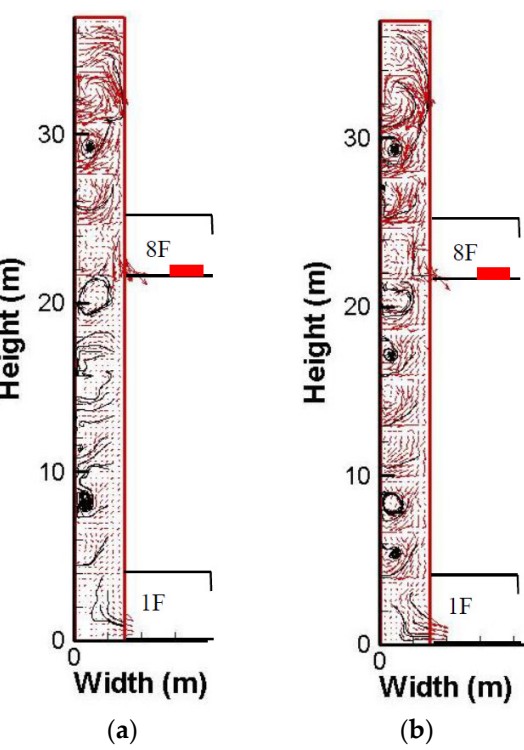

**Figure 9.** Velocity vector graph at 4 m/s: (**a**) 120 s and (**b**) 390 s.

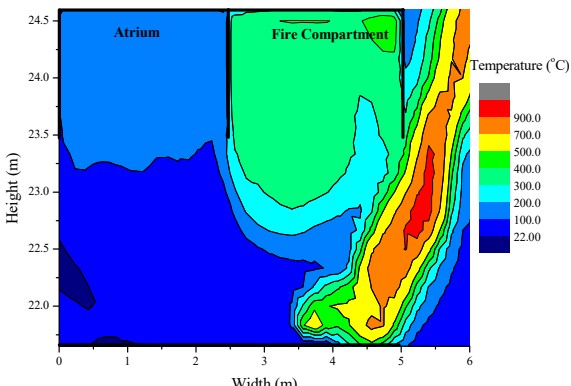

**Figure 10.** Temperature distributions in the atrium and fire compartment at 4 m/s and 120 s.

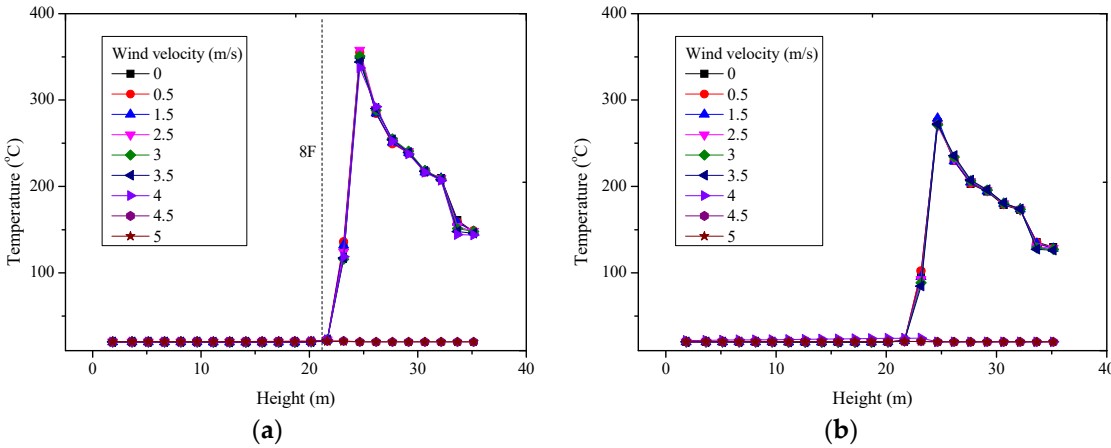

**Figure 11.** Vertical temperature distributions in the staircase under different wind velocities: (**a**) 4 MW and (**b**) 3 MW.

### 3.3. Smoke Back-Layering Distance under the External Wind

Smoke behaviors in all cases are shown in Table 2. It is shown that the smoke moves upward when the external wind velocity is less than 3 m/s regardless of the open stair door location and HRR. With the increasing wind velocity, three kinds of smoke behaviors are identified: first downward then upward moving smoke, downward moving smoke, and no smoke inside the staircase. The transition wind velocity between upward moving smoke and downward moving smoke is about 3.5–4 m/s at 2 MW and about 4–4.5 m/s at 4 MW. The transition wind velocity between downward moving smoke and no smoke inside the staircase is about 4–5 m/s at 2 MW and about 4.5–5.5 m/s at 4 MW. It can be concluded that the transition wind velocity of the three behaviors increased slightly with the increasing HRR.

When the smoke moves downward and flows out from the open stair door below the fire floor, the back-layering distance is the vertical height between the fire floor and the open stair door. Then, we will focus on the longest back-layering distance in cases with first downward then upward moving smoke. The back-layering distance is determined temperature variations at different height. Tanaka [6] stated that the temperature would rise suddenly when the front of smoke plume reaches the thermocouple at a given height. Take the case with open stair door on the fifth floor at 3.5 m/s as an example, as shown in Figure 12. From Figure 12a, the lowest thermocouple with a sudden temperature rise is at 20.15 m. The temperature gets a peak value at 44 s. Then temperature field of the staircase at 44 s can be obtained, as shown in Figure 12b. The temperature of the downward smoke plume front is set as 22 °C, which is determined based on the visual smoke in the smokeview program. The back-layering distance is the height between the fire floor and the downward smoke plume front, as shown in Figure 12b.

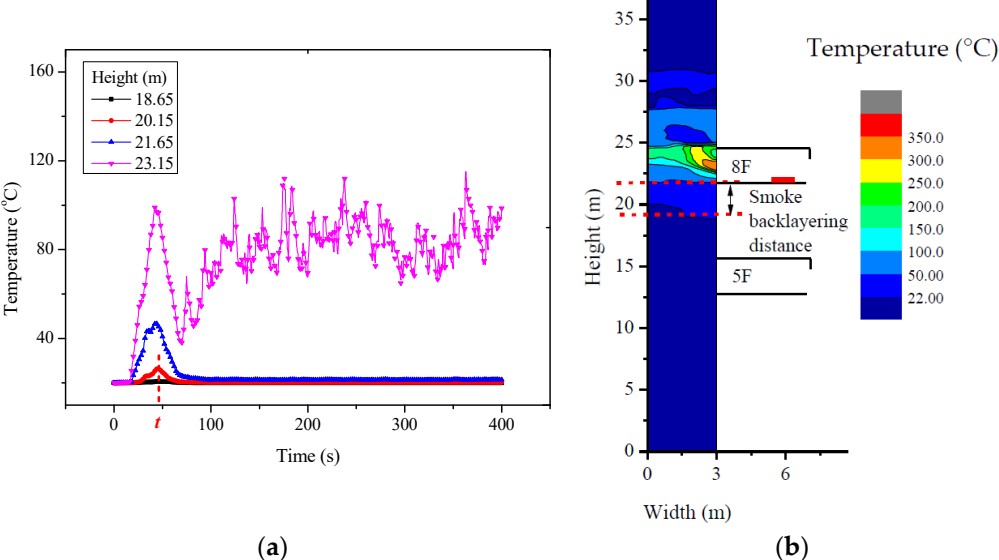

**Figure 12.** Method to determine the smoke back-layering distance. (**a**) Time history of the temperatures. (**b**) Temperature field at 44 s.

**Table 2.** Smoke behaviors in all cases.

| Wind Velocity (m/s) | | 0, 0.5, 1.5, 2.5 | 3 | 3.5 | 4 | 4.5 | 5 | 5.5 |
|---|---|---|---|---|---|---|---|---|
| Opening Location | HRR(MW) | | | | | | | |
| 1F | 2 | ↑ | T | ↓↑ | ↓ | N | N | - |
| | 3 | ↑ | T | ↓↑ | ↓ | N | N | - |
| | 4 | ↑ | T | ↓↑ | ↓↑ | ↓ | N | - |
| 3F | 2 | ↑ | T | ↓↑ | ↓ | N | N | - |
| | 3 | ↑ | T | ↓↑ | ↓ | ↓ | N | - |
| | 4 | ↑ | T | ↓↑ | ↓↑ | ↓ | N | - |
| 5F | 2 | ↑ | T | ↓↑ | ↓ | ↓ | N | - |
| | 3 | ↑ | T | ↓↑ | ↓↑ | ↓ | N | - |
| | 4 | ↑ | T | ↓↑ | ↓↑ | ↓ | N | - |
| 7F | 2 | ↑ | T | ↓↑ | ↓ | ↓ | N | - |
| | 3 | ↑ | T | ↓↑ | ↓↑ | ↓ | ↓ | N |
| | 4 | ↑ | T | ↓↑ | ↓↑ | ↓ | ↓ | N |

↑ upward moving smoke, ↓ downward moving smoke, ↓↑ first downward then upward moving smoke. T, critical transition condition of the upward moving smoke; N, no smoke inside the staircase.

The longest back-layering distance in cases with first downward then upward moving smoke can be seen in Figure 13. From Figure 13, the back-layering distance time and distance increase with the increasing open stair door location and the wind velocity. However, it decreases with the increasing HRR. Tanaka et al. [6] proposed a non-dimensional travel time $\tau$ to research the rise time of plume front inside the staircase, which can be expressed as

$$\tau = t\sqrt{\frac{g}{D}}\left[\frac{Q}{\rho_a T_a C_p \sqrt{g} D^{2.5}}\right]^{1/3} \tag{9}$$

where $D$ is the characteristic length of the staircase, $g$ is the acceleration due to gravity (m/s$^2$), $\rho_a$ is the density of air (kg/m$^3$), $T_a$ is the temperature of air (k), and $C_p$ is specific heat of air at constant pressure (kJ/kg K). Then, the relation between dimensionless rise-time of fire plume front and dimensionless height was established. We try to use the same method to study the smoke back-layering time and distance. Dimensionless back-layering time $\tau'$ can be expressed as

$$\tau' = \frac{t}{t_o}\left[\frac{Q}{\rho_a T_a C_p \sqrt{g} L^{2.5}}\right]^{1/3} \tag{10}$$

where $t_o$ is the time when the smoke plume front reaches the top of the staircase without external wind, and $L$ is the smoke back-layering distance. The dimensionless smoke back-layering distance, $L^*$, can be expressed as

$$L^* = L/H \tag{11}$$

where $H$ is vertical height between the fire floor and the top of the staircase. The relation between $\tau'$ and $L^*$ can be seen in Figure 14. As shown in Figure 14, the dimensionless smoke back-layering distance is linearly proportional to the new dimensionless back-layering time in double logarithmic coordinates. However, the wind effect is neglected. The downward moving smoke inside the staircase is mainly resulted from the external wind and influenced by the HRR of the fire. Then the correlation will be modified to make it more reasonable.

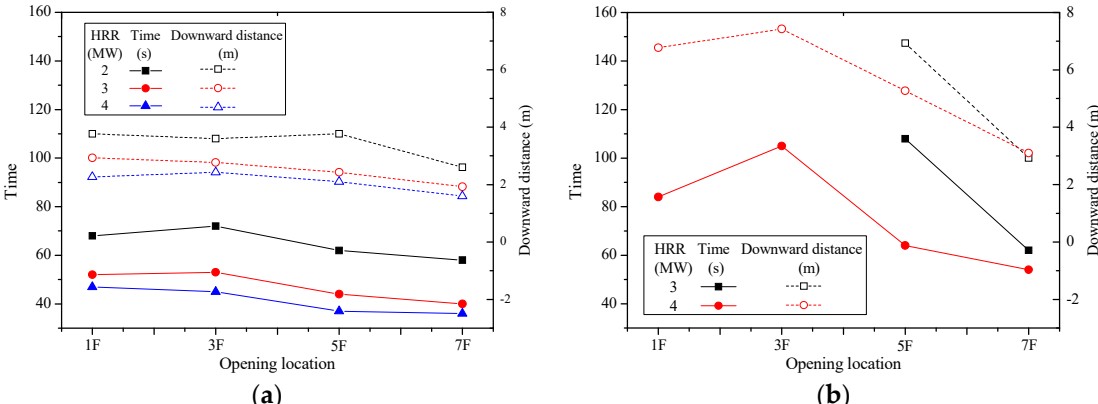

**Figure 13.** Longest smoke back-layering distance with different open stair doors and wind velocities. (**a**) 3.5 m/s. (**b**) 4 m/s.

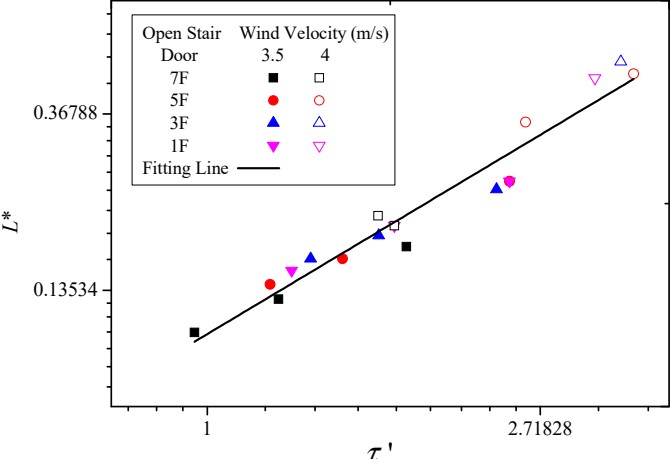

**Figure 14.** Dimensionless back-layering distance $L^*$ versus dimensionless back-layering time $\tau'$.

The smoke back-layering phenomenon usually occurred in the narrow and long subway tunnel. The back-layering distance is important for the ventilation system design. When the longitudinal velocity is high enough, the smoke back-layering upstream of the fire will be prevented. Prediction correlations for the back-layering distance showed that the dimensionless back-layering distance was proportional to the dimensionless HRR and inversely proportional to the dimensionless ventilation velocity [25–27]. Based on the studies for the smoke back-layering phenomenon in the subway tunnel, the dimensionless back-layering time can be modified as

$$\tau_{\mathrm{mod}} = \frac{t}{t_o} \left[ \frac{Q}{\rho_a T_a C_p \sqrt{g} L^{2.5}} \right]^{1/3} \left( \frac{v^2}{gL} \right)^{\alpha} \tag{12}$$

Correlating experimental data with the modified dimensionless back-layering time and dimensionless smoke back-layering distance, we obtain Figure 15. It can be seen that, in double logarithmic coordinates, the dimensionless smoke back-layering distance are linear related to dimensionless back-layering time when $\alpha = -1$.

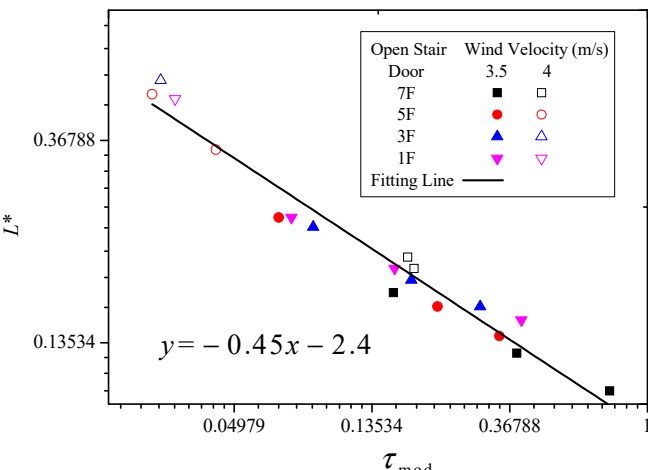

**Figure 15.** Dimensionless back-layering distance $L^*$ versus modified dimensionless back-layering time $\tau_{\mathrm{mod}}$.

## 4. Conclusions

In this study, a set of numerical simulations is conducted to investigate the combined external wind and stack effect on smoke back-layering phenomenon in the staircase with open stair doors below the fire floor. The major conclusions are summarized as follows:

(1) When the external wind is absent, downward moving smoke occurred, driven by pressure difference with closed stair doors below the fire floor.

(2) When the stair doors open below the fire floor, four smoke behaviors are identified with the increasing wind velocity: upward moving smoke, first downward then upward moving smoke, downward moving smoke, and no smoke inside the staircase.

(3) Smoke behaviors are directly related to the flame tilt direction. Smoke moves upward when the flame tilts to the staircase and moves downward when the flame tilts to the opposite direction.

(4) The back-layering distance with the first downward then upward moving smoke increases with the increasing open stair door location and wind velocity. A modified correlation is established to predict the smoke back-layering distance and time.

**Author Contributions:** Writing—original draft preparation, M.L.; formal analysis, L.W. and J.C.; software, Z.M. and S.L. All authors have read and agreed to the published version of the manuscript.

**Funding:** This research was funded by National Natural Science Foundation of China (NSFC) under Grant No. 52006075, opening fund of the State Key Laboratory of Fire Science under Grant No. HZ2019-KF06 and the scientific research funds of Huaqiao University under Grant No. 605-50Y18057.

**Data Availability Statement:** Not available.

**Acknowledgments:** This work was supported by National Natural Science Foundation of China (NSFC) under Grant No. 52006075, opening fund of the State Key Laboratory of Fire Science under Grant No. HZ2019-KF06 and the scientific research funds of Huaqiao University under Grant No. 605-50Y18057.

**Conflicts of Interest:** The authors declare no conflict of interest.

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
