# Peer review of "Smoke Back-Layering Phenomenon under the Combined External Wind and Stack Effects in a Staircase"

_applsci, doi:10.3390/app12031469_

Round 1

Reviewer 1 Report

This paper investigated the smoke movement patterns under different combination conditions of open stair door location, heat release rate of fires, and external wind velocities. Its findings are useful to protect human safety under high-rise building fires. This manuscript is generally well-written. However, it could be improved further based on the following comments:

  1. Line 33: Please change ‘th e’ to ‘the’.
  2. Line 51: Please change ‘observ’ to ‘observe’.
  3. Lines 54-62: From these sentences, I cannot see what’s new in this study relative to Li et al. [2]. You may need refine these sentences to highlight the challenges and significance of this study, rather than highlight the findings from Li et al. [2].
  4. Line 68: Full name of FDS?
  5. Chapter 2: What’s your contribution to this model? Just use the existing model and building structure (Figure 1)? What is the creativity of this paper relative to pervious publications? Please clarify in the manuscript.
  6. Line 93: Why do you set the building height as 12-story? Will the conclusion change if the building height change?
  7. Lines 93-105: How do you decide these parameters in the building model or is it a fixed building model in the FDS model?
  8. Table 1: What if stairs doors at different stories are open at the same time? Have you simulated this situation?
  9. Figure 2: I suggest to explain the x- and y-axis labels in the figure caption, because I am not clear about the meanings of r, H and ΔT and T.
  10. Line 144: It is not clear what does ‘each other’ refer to?
  11. Line 150: Does 300-400s refer to the time from the start of fire? Please clarify here.
  12. Lines 154/157: Please check the temperature unit through the manuscript.
  13. Figures 5 and others: I suggest to add unit and name for the color legend. In addition, it is not clear that Figure 5 shows the ‘Pressure difference distribution’ or ‘Pressure difference’ because the descriptions are not consistent between Figure 5 caption and context.
  14. Lines 144-145, 182-183: If the conclusion that ‘smoke behaviors….are similar with different …’ has been verified by yourself or other publications? Please clarify here.
  15. Line 225: 4.5m/s or 5m/s? It is not consistent between figure and context. Please check through the manuscript to make all the wind velocities correct!
  16. Lines 229-230: This sentence has been repeated with line 150.
  17. Table 2: What is the meaning of empty cells in this table?
  18. Figures 14/15: Since the figure should be self-explanatory, I suggest to add more explanations in the captions of this figure and other figures.
  19. Line 295: τ' is not explained.

Reviewer 2 Report

In this paper, a numerical analysis was conducted to investigate the combined external wind and stack effect on smoke back-layering phenomenon in a staircase.

It seems that various conditions are well examined in accordance with the location of open stair doors relative to the fire floor.

A few questions, however, need to be replied or supplemented as follows.

1) Definition of 'Smoke back-layering' should be presented in the text.

2) Why is not there analysis about the condition for open stair doors above the fire floor? What results do you expect in this condition?

3) Why was wind velocity 5.5m/s applied to only 2 cases(7th open stair door)?

4) Wind velocity in Fig. 6d doesn't accord with the velocity on the text(225th row).

5) Why do you think the transition wind velocity of the 3 behaviors increased slightly with the increasing HRR?
